# Investigation of the Performance and Emission Characteristics of Diesel Engine Fueled with Biogas-Diesel Dual Fuel

**Melkamu Genet Leykun** [1,*] **and Menelik Walle Mekonen** [2]

1 Faculty of Mechanical Engineering, Arbaminch University, Arbaminch 4400, Ethiopia
2 Department of Motor Vehicle Engineering, Defense University College of Engineering, Debrezeit 1041, Ethiopia; menelikiitg@gmail.com
* Correspondence: m8genet@gmail.com

**Abstract:** Due to the popularity of diesel engines, utilization of fossil fuel has increased. However, fossil fuel resources are depleting and their prices are increasing day by day. Additionally, the emissions from the burning of petroleum-derived fuel is harming the global environment. This work covers the performance and emission parameters of a biogas-diesel dual-fuel mode diesel engine and compared them to baseline diesel. The experiment was conducted on a single-cylinder and four-stroke DI diesel engine with a maximum power output of 2.2 kW by varying engine load at a constant speed of 1500 RPM. The diesel was injected as factory setup, whereas biogas mixes with air and then delivered to the combustion chamber through intake manifold at various flow rates of 2, 4, and 6 L/min. At 2 L/min flow rate of biogas, the results were found to have better performance and lower emission, than that of the other flow; with an average reduction in BTE, HC, and NOx by 11.19, 0.52, and 19.91%, respectively, and an average increment in BSFC, CO, and $CO_2$ by 11.81, 1.05, and 12.8%, respectively, as compared to diesel. The diesel replacement ratio was varied from 19.56 to 7.61% at zero engine load and 80% engine load with biogas energy share of 39.6 and 16.59%, respectively.

**Keywords:** alternative fuel; biogas; diesel engine; dual-fuel; emission; performance

## 1. Introduction

The global energy demand is rising swiftly due to the increase in population and the continuous growth of industrialization, which leads to an increase in the usage of fossil fuels. It supplies above 80% for global energy usage and around 95% energy for the transport sector internationally as reported by Singh et al. [1]. However, fossil fuels are rapidly depleting, have increasing prices, and also present a global challenge for easy accessibility, because the known conventional petroleum-derived fuel reservoirs are in countries in politically unstable regions as described by Turco et al. [2] and P.A. Singh et al. [3]. On the other hand, the combustion of fossil fuel hurts the environment by creating acid rain, increasing environmental warming, rises in greenhouse gas concentrations, etc. in the case of diesel engines, due to it releasing more nitrogen oxides (NOx) and particulate matter (PM) as compared to a petrol engine, as stated by Neeranjan and Layek [4].

Hence, CI engines are increased in their popularity due to their best trusty and competent internal combustion engine, Yoon and Lee [5]. Furthermore, its exhaust emissions are part of environmental pollutants, Karim and Sulaiman [6]. However, the positive achievements in terms of efficiency associated with it are overshadowed by its high emission drawbacks. Due to these reasons on the demand as well as the effect of petroleum-derived fuels, searching and utilizing an alternate fuel for diesel engines have become a prominent research area.

Alternative fuels release a smaller amount of emissions when used in internal combustion engines as compared to conventional fuels, Bhuimbar and Kumarappa [7]. Thus, biogas is a good gaseous alternative fuel to replace petroleum-derived fuels, Ayade and

Latey [8], and releases less emission, Salve et al. [9], so that it is economical as well as suitable to the environment [7]. Biogas can be produced using anaerobic digestion through the fermentation of different biodegradable materials, Mustafi and Agarwal [10] and Prajapati et al. [11], with a composition of methane CH4 (40–75%), carbon dioxide (15–60%), and other trace gases, Bharathiraja et al. [12]. The methane percentage in raw biogas can be also improved to natural gas standards and becomes bio-methane, Mustafi and Agarwal [10] and Prajapati et al. [11]. It is not only an abundant resource of renewable energy for countries that face an economic challenge for importing conventional fuels, but also higher economic countries where environmental gas emission is a major issue, Prajapati et al. [13].

As recent studies have shown, there is an emerging domestic energy source (i.e., biogas) in Ethiopia, Kamp and Forn [14], led by the Ethiopian National Biogas Program (NBPE). It may be produced in huge amounts from municipality abattoir wastes, Solomon et al. [15], and correspondingly improves environmental waste management. However, a large amount of solid waste disposal from a large number of livestock resources in Ethiopia increases methane emissions, Idriss and Mekonnen [16] and Yonas et al. [17]. One alternative way to minimize methane emissions is using solid waste disposal (which causes methane emission) to use energy sources, i.e., using it for producing biogas. This promotes the utilization of biogas for automobiles, industries, stationary CI engines like generators for different organizations, and for household applications.

The higher thermal efficiency of biogas, Prajapati et al. [13], and the higher compression ratio property of diesel engines, Yoon and Lee [5], provides the compatible application of biogas as a fuel for diesel engines. Additionally, higher self-ignition temperature properties of biogas allow it to run the engine at lean mixtures by increasing performance and reducing harmful released gasses, Barik and Murugan [18]. However, it cannot be used to run a diesel engine directly, Prajapati et al. [11], and also cannot be applied alone, which requires a separate fuel delivery system, Salve et al. [9] and Mustafi and Agarwal [19]. Biogas-diesel fuel engine consists of a primary fuel (biogas) and a pilot fuel used for ignition (diesel), Prajapati et al. [11].

From this perspective, the purpose of this research is to look into the performance and emission characteristics of raw biogas in diesel engines, as well as to promote the use of biogas in developing countries due to its readily available source, environmentally friendly nature, and minimizing the cost of importing conventional petrol-derived fuels. Furthermore, the majority of petro-derived fuel reserver countries are political hotspots. In this paper the qualities of CI engines were studied when they were driven on baseline diesel and biogas-diesel dual fuel. These properties are: brake thermal efficiency, brake specific fuel consumption, biogas energy share, and emissions, such as carbon monoxide, carbon dioxide, unburned hydrocarbons, and nitrogen oxides. A single-cylinder computer-controlled test bench diesel engine (located at AASTU in Addis Ababa, Ethiopia) was used to investigate the influence of biogas flow rate on engine performance and emission characteristics while varying engine load and comparing it to neat diesel fuel.

## 2. Methodology

A well-designed methodology was used to address this work. To complete this study, some procedures were used in tandem. These included determining the composition of biogas, preparing fuels (collecting biogas from digester outlets and purchasing diesel fuel from fuel stations), preparing the experimental setup of the test engine, analyzing experimental results for each mode of operation, and finally comparing performance and emissions to that of baseline diesel fuel.

A Geotech biogas analyzer (from Addis Ababa University, Ethiopia) was used to analyze the composition of raw biogas obtained from cow dung; all of its constituents are given in Table 1 below, and the raw biogas sample was taken using a Glucose bag. Since methane is the only combustible gas, Table 1 shows that over 55.5 percent of the material is non-combustible and considered impurities.

**Table 1.** Composition of raw biogas.

| Contents | Percentage (% by Vol.) |
|----------|------------------------|
| $CH_4$ | 44.5 |
| $CO_2$ | 10.7 |
| $O_2$ | 8.7 |
| $H_2O$ | 0 |
| Traces gases | 36.1 |

After determining the composition of biogas, it was collected from the digester outlet and compressed into the tire tube by using an ordinary compressor. The properties of raw biogas and commercial diesel fuel used in this study are shown in Table 2 below.

**Table 2.** Fuel properties, Bouguessa et al. [20].

| Properties | Diesel | Biogas |
|------------|--------|--------|
| Lower heating value (MJ/kg) | 42 | 20.3958 * |
| Density (kg/m$^3$) | 840 | 1.12 * |
| Auto ignition temperature (°C) | 280 | 650 |
| Stoichiometric air-fuel ratio | 14.60 | 17 |
| Cetane number | 49 | - |
| Octane number | - | 130 |
| Laminar burning velocity (m/s) | 0.5 | 0.2 |

* calculated value.

As shown in Figure 1, the various experiments were carried out after the venturi gas mixer was mounted at the air intake manifold following the airflow sensor. After the engine starts in diesel fuel mode, the biogas flow control valve slowly opened. The vacuum formed at the throat in diesel-only mode sucks the gas from its source to the mixer. As a result, the amount of airflow reaching the engine cylinder reduces, and biogas takes its place. Frequently, the engine will switch to dual-fuel mode. Due to the engine injecting a pilot fuel to ignite the charge proportional to the amount of air in, the amount of diesel fuel injected for ignition reduces (i.e., due to a decrease in the amount of air entered). Three variable volume flow rates of biogas with diesel fuel in dual fuel mode were used in the experiment. Each trial was carried out three times on different days, with sufficient care taken to load the engine accurately at each phase of the load and to maintain steady ambient conditions.

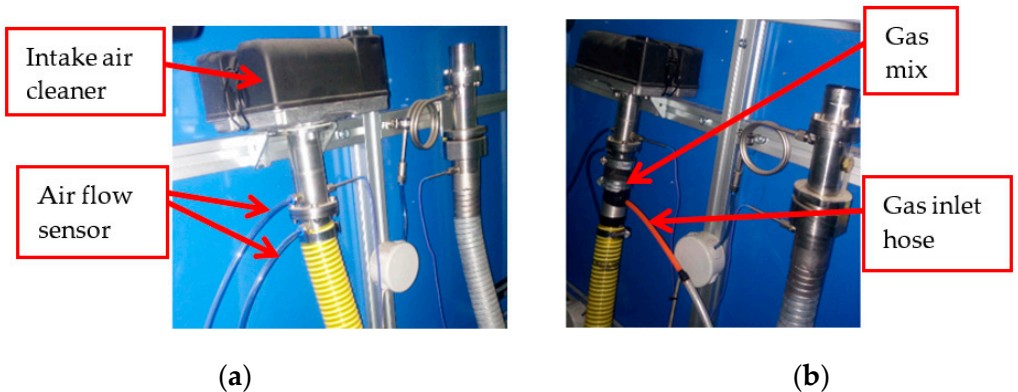

(**a**)                                                                (**b**)

**Figure 1.** Installation of venturi gas mixer at the air intake manifold. (**a**) Before installing gas mixer, (**b**) after installing a gas mixer.

All of the tests were conducted after the engine had attained the operational temperature and at a constant engine speed of 1500 RPM that was programmed into the computer. The load fluctuated from zero to 80% in a ten percent step-up, and as the load changed, so

did the brake power and brake torque, which was shown on the data acquisition computer. From these results, BTE and BSFC were calculated by the formula, Gurung et al. [21]:

$$\eta_{bth}(\%) = \frac{P_b}{\dot{m}_d \times LHV_d} \times 100 \tag{1}$$

where,

$$\eta_{bth} = \text{brake thermal efficiency,}$$

$$P_b = \text{brake power in kW,}$$

$$\dot{m}_d = \text{mass flow rate of diesel in pure diesel mode in} \frac{kg}{h},$$

$$LHV_d = \text{lower heating value of diesel in kJ/kg.}$$

While in the case of dual-fuel mode, brake thermal efficiency is given by the following formula, Bouguessa et al. [20] and Gurung et al. [21]:

$$\eta_{bth}(\%) = \frac{P_b}{\dot{m}_{bg} \times LHV_{bg} + \dot{m}_{dd} \times LHV_d} \times 100 \tag{2}$$

where,

$$\dot{m}_{dd} = \text{mass flow rate of diesel fuel in a dual} - \text{mode in kg/h}$$

$$\dot{m}_{bg} = \text{mass flow rate of biogas in kg/h}$$

$$LHV_{bg} = \text{lower heating value of biogas in kJ/kg}$$

When the testing engine run in diesel fuel mode, bsfc is calculated as, Ambarita H. [22]:

$$bsfc = \frac{\dot{m}_d}{P_b} \tag{3}$$

where;

$$bsfc = \text{brake specific fuel consumption in kg/kWh}$$

While in the case of diesel engine operated in dual fuel mode, bsfc is given by the following formula [22]:

$$bsfc = \frac{\dot{m}_{dd} + \dot{m}_{bg}}{P_b} \tag{4}$$

The biogas flow control valve should be slowly opened after restarting the engine with diesel fuel to provide biogas from the tire tube and operate the engine in dual fuel mode. Rotameter controlled and changed the flow rate, as seen in Figure 2a below and its measuring range is shown in Table 3 below. Hence, according to the baseline test matrix provided in Table 4, the diesel engine is first tested using diesel fuel to get baseline data for each loading scenario. Then, as maintained by the dual-fuel mode test matrix presented below in Table 5, the flow rate of biogas in dual fuel mode was demonstrated at 2, 4, and 6 L/min. The results were recorded for each load situation using diesel (D) fuel, D + BG@2 L/min flow rate, D + BG@4 L/min flow rate, and D + BG@6 L/min flow rate. As illustrated in Figure 2b, the diesel fuel flow rate was also measured using the fuel burette.

**Table 3.** Rotameter measurement range.

| Device | Measuring Ranges | Resolution | Accuracy | Relative Errors (%) |
|--------|------------------|------------|----------|---------------------|
| Rotameter | 0–50 L/min | ±0.1 L/min | ±5 L/min | ±1 |

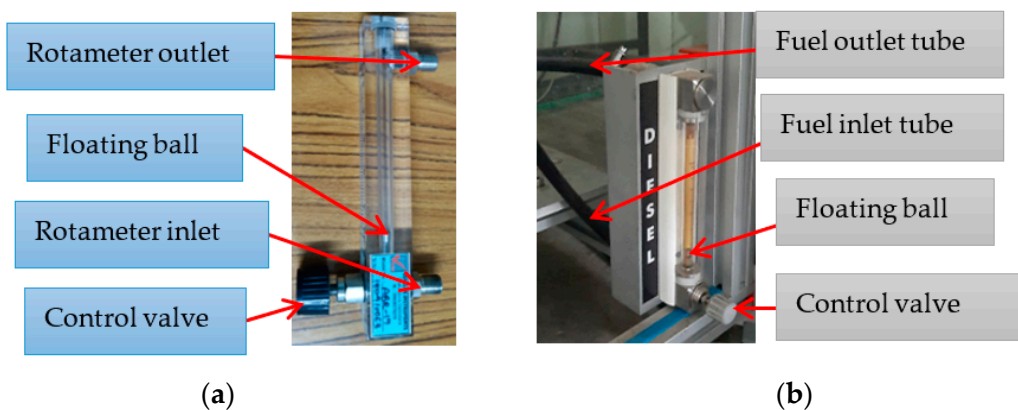

**Figure 2.** Flowmeters. (**a**) Rotameter, (**b**) burette.

**Table 4.** Baseline data experimental test matrix.

| Mode of Operation | Engine Operating Parameters | | Evaluating Engine Parameters |
|---|---|---|---|
| | Fixed CR, IP, and IT | Load | |
| Diesel fuel mode | Set fixed as factories recommender | 0% 10% 20% 30% 40% 50% 60% 70% 80% | a. Performance parameters<br>i. Brake Torque ($T_b$)<br>ii. Brake Power ($P_b$)<br>iii. Brake specific fuel consumption (BSFC)<br>iv. Brake thermal efficiency (BTE)<br>b. Emission Parameters<br>i. CO<br>ii. HC<br>iii. $CO_2$<br>iv. NOx |

**Table 5.** Experimental test matrix for dual fuel mode operation.

| Mode of Operation | Engine Operating Parameters | | Biogas Flow Rate | Evaluating Engine Parameters |
|---|---|---|---|---|
| | Fixed CR, IP, and IT | Load | | |
| Dual fuel mode$(Diesel-Biogas) | Set fixed as factories recommender | 0% | 2 L/min, 4 L/min, 6 L/min | a. Performance parameters<br>i. Brake Torque ($T_b$)<br>ii. Brake power ($P_b$)<br>iii. Brake specific fuel consumption (BSFC)<br>iv. Brake thermal efficiency (BTE)<br>b. Emission Parameters<br>i. CO<br>ii. HC<br>iii. $CO_2$<br>iv. NOx |
| | | 10% | 2 L/min, 4 L/min, 6 L/min | |
| | | 20% | 2 L/min, 4 L/min, 6 L/min | |
| | | 30% | 2 L/min, 4 L/min, 6 L/min | |
| | | 40% | 2 L/min, 4 L/min, 6 L/min | |
| | | 50% | 2 L/min, 4 L/min, 6 L/min | |
| | | 60% | 2 L/min, 4 L/min, 6 L/min | |
| | | 70% | 2 L/min, 4 L/min, 6 L/min | |
| | | 80% | 2 L/min, 4 L/min, 6 L/min | |

*Experimental Test Matrix*

An experimental test matrix is the number of tests that were being executed and guidance for experimental work. Baseline and dual fuel (DF) mode experimental test matrices are listed below.

### 3. Experimental Setup

The experimental test diesel engine is a single-cylinder, four-stroke, naturally aspirated, direct injection, and computer-controlled test bench diesel engine TBMC3-02. It is equipped with an eddy current asynchronous motor dynamometer (AM-1) for loading the engine. A factory engine electronic control unit (ECU) controlled the timing and duration of diesel injection, and none of the programmed parameters were changed. Furthermore, all of the sensors on the engine were completely linked to the ECU using factory settings. Providing a compression ratio of 21:1, a nozzle opening pressure of 210 bar, and a typical injection timing of 23° bTDC in the engine. The schematic and photographic representation of the experimental setup is illustrated in Figure 3. The setup enables the measurement of the performance parameters and emission constituents. The engine performance parameters are BSFC and BTE, whereas exhaust emission levels are measured using an SV-50 automotive emission analyzer, which measures CO and $CO_2$ constituents in %Vol, whereas hydrocarbon and $NO_X$ are in ppm and its measuring range is shown in Table 6 below. The experimental investigation was done on three different volume flow rates of biogas (2, 4, and 6 L/min) with diesel. The technical specification of a test engine is shown in Table 6 below and Table 7 illustrates SV-50 automotive emission analyzer measuring range.

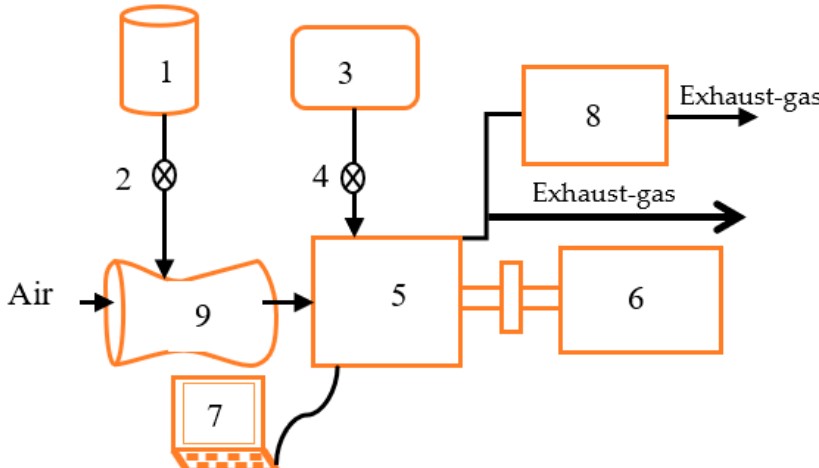

**Figure 3.** Schematic experimental setup of the test engine. 1 Compressed biogas. 2 Rotameter. 3 diesel fuel tank. 4 Burette. 5 Engine (TBMC3-02). 6 Dynamometer. 7 Data acquisition computer. 8 Exhaust gas analyzer. 9 Air-biogas mixer.

**Table 6.** Test engine TBMC3-02 technical specification.

| Engine model | TBMC3-02 |
|---|---|
| Type of engine | Compression ignition |
| Fuel | Diesel |
| Number of cylinders | 1 |
| Type of air intake | Naturally aspirated |
| Cooling type | Air |
| Compression ratio, CR | 21:1 |
| Bore and Stroke diameter | 69 mm by 60 mm |
| No. of stroke | 4 stroke |
| Torque | 7.4 N.m |
| Engine power | 2.2 kW |
| Air inlet diameter (D) | 36 mm |

**Table 7.** Automobile Exhaust Gas Analyzer SV-50 measuring range.

| Species | Measurement Range | Resolution | Allowed Error | Relative Error |
|---|---|---|---|---|
| HC | 0–1000 ppm | 1 ppm | ±12 ppm | ±5% |
| CO | 0–10.0 in % Vol. | 0.01% | ±0.06% | ±5% |
| $CO_2$ | 0–20.0 in % Vol. | 0.1% | ±0.5% | ±5% |
| NOx | 0–800 ppm | 1 ppm | ±10 ppm | ±5 |

## 4. Result and Discussion

This section analyzes and discusses biogas energy sharing and performance parameters such as BTE and BSFC after proceeding with the experiment using diesel as a baseline and diesel with biogas in a dual-mode. Biogas energy share (BGES) indicates whether biogas is a high-density or low-density fuel by the amount of energy it contributes to the total energy required in dual fuel mode. According to Deheri et al. [23], brake thermal efficiency (BTE) is a basic engine performance characteristic that assesses the engine's ability to transfer theenergy held in the fuel to mechanical energy during combustion. The fuel efficiency of any engine that consumes fuels and creates rotational power output is measured by brake specific fuel consumption (BSFC), and its value reflects how efficiently the engine transforms fuel given into useable brake power (work), B. Ashok and K. Nanthagopal [24]. In this section, emission parameters were also recorded and discussed, as indicated below.

### 4.1. Mass Fraction of Biogas and Biogas Energy Share

a. Mass fraction of biogas

Table 6 shows the mass fraction of biogas for each mode of operation. The biogas flow was set fixed for each mode of operation. However, when we look at the experiment results, which are summarized in Table 8, we can see that the diesel fuel flow rate increases to maintain the required power when the engine was imposed a higher thermal load. As a result, the biogas energy share decreases at higher engine loads, as discussed further below in Section 4.1b.

**Table 8.** Mass fraction of biogas.

| Mode of Operation | Load (%) | $\dot{m}_{dd}$ (kg/h) | $\dot{m}_{bg}$ (kg/h) | BGES (%) |
|---|---|---|---|---|
| Dual-mode Diesel + Biogas at 2 L/min flow rate | 0 | 0.0995 | 0.1344 | 39.6 |
| | 20 | 0.146 | 0.1344 | 30.88 |
| | 40 | 0.195 | 0.1344 | 25.07 |
| | 60 | 0.251 | 0.1344 | 20.63 |
| | 80 | 0.328 | 0.1344 | 16.59 |
| Dual-mode Diesel + Biogas at 4 L/min flow rate | 0 | 0.0897 | 0.2688 | 59.26 |
| | 20 | 0.137 | 0.2688 | 48.78 |
| | 40 | 0.185 | 0.2688 | 41.36 |
| | 60 | 0.235 | 0.2688 | 35.70 |
| | 80 | 0.314 | 0.2688 | 29.35 |
| Dual-mode Diesel + Biogas at 6 L/min flow rate | 0 | 0.084 | 0.4032 | 70.08 |
| | 20 | 0.129 | 0.4032 | 60.25 |
| | 40 | 0.175 | 0.4032 | 52.86 |
| | 60 | 0.215 | 0.4032 | 47.69 |
| | 80 | 0.306 | 0.4032 | 39.02 |

b Biogas energy share (BGES)

The discrepancy of biogas energy contribution through engine load is shown in Figure 4 and it is noticed that higher energy contribution of biogas at low loads, while it decreases with increasing load, for all the three modes of operation. This is because an increment of engine load requires higher density fuel (diesel) to compensate for the higher

executed thermal load on the engine and since the amount of biogas flow was set fixed for each dual-fuel operation, whereas diesel varies. Whereas, as the biogas flow rate increases, its energy share increases due to a decrease in pilot fuel compared with low biogas flow rate operations. However, for an increment of load, the consumption of baseline fuel also goes up for all operations. Hence, for 2 L/min (0.134 kg/h), 4 L/min (0.268 kg/h), and 6 L/min (0.4 kg/h) flow of biogas, an average biogas energy share of 24.98, 40.95, 52.32%, and average diesel replacement ratio of 13.41%, 18.16 and 23.29 was noticed, respectively. A maximum biogas energy share of 65.82% with a maximum diesel fuel replacement ratio of 30.53% was observed at 6 L/min biogas flow rate on a low engine load of 10%. Generally, increasing in biogas flow rate increases its energy share as well as an increase in diesel replacement ratio.

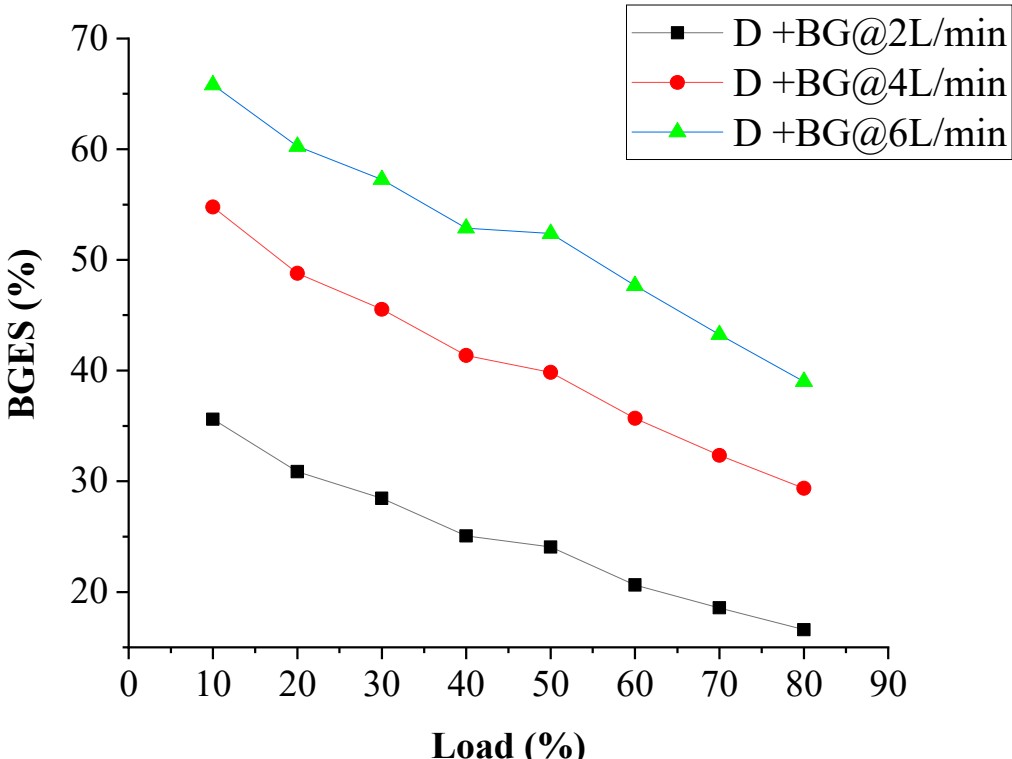

**Figure 4.** Biogas energy share.

*4.2. Engine Performance*

a.     Brake thermal efficiency (BTE)

Figure 5 shows the variation of BTE and for all modes of operation, it increases as the engine load increases, but is noticed lower in the case of biogas-diesel fuel operation as compared to the baseline. This is because of the high resistance to auto-ignition properties and the high octane number of biogas. These properties of biogas decrease the combustion temperature. This leads to the lower energy conversion efficiency of the diesel-biogas mixed fuel as compared to the baseline and thereby decreases BTE. Similarly, due to raw biogas containing more $CO_2$ content, $CO_2$ restricts the rapid burning of the mixture during combustion. This affects not only the burning speed, but also causes a reduction in flame propagation, which then results lower in energy conversion efficiency during diesel-biogas fuel mode operation. A similar trend was reported by Rosha et al. [25]. In addition to this, in biogas-diesel mode operations, as biogas flow rate increases, BTE decreases. This is due to an increase in the induction of biogas, which leads to further decreases in the flame propagation speed and resulted in lower BTE. Generally, an average BTE reduction of D

+ BG@2 L/min, D + BG@4 L/min, and D + BG@6 L/min flow rate mode obtained 11.19, 18.45, and 25.72%, respectively, as compared to diesel mode.

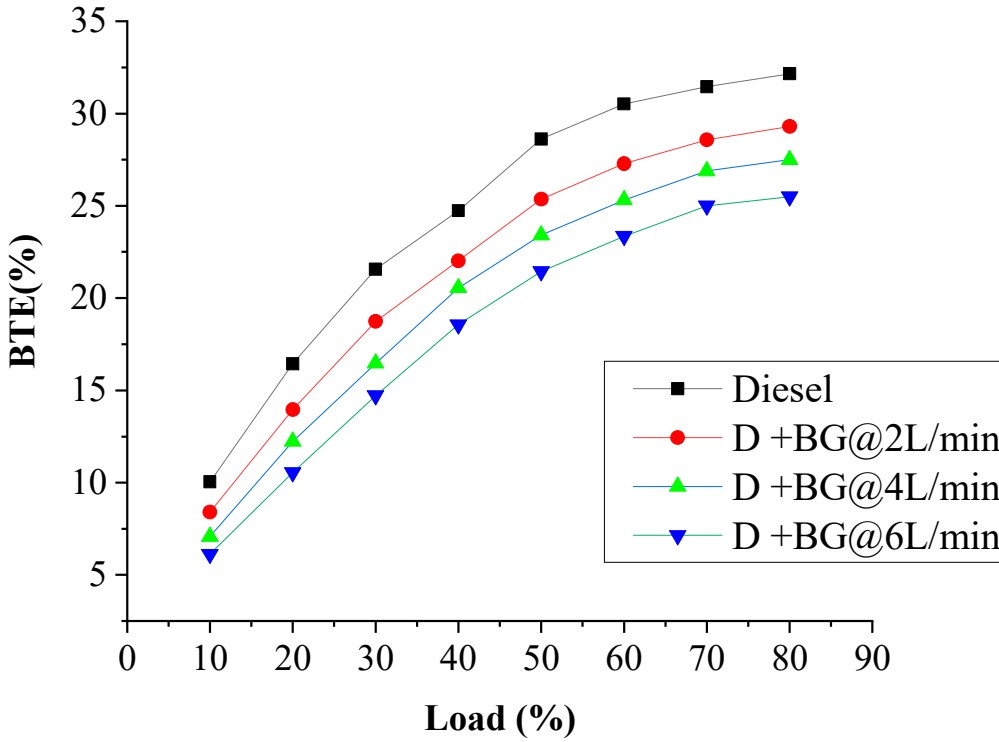

**Figure 5.** Variation of BTE with respect to engine load.

b       Brake-specific fuel consumption (bsfc)

Brake-specific fuel consumption depends on the heating value of the fuel, Sandalc et al. [26]. Figure 6 shows brake-specific fuel consumption for both baseline and biogas-diesel fuel. At a lower engine load, higher brake-specific fuel consumption is noticed for both modes. This is due to lower output power at a lower load. However, it was found to be lesser at high engine load for both modes of operations because of an increase in combustion rate because of high AFR and high burning temperature. From Figure 6, it is observed that supplying biogas leads to an increase in fuel consumption as compared to baseline throughout the load range. This is the reason for low energy density in addition to the slow-burning of raw biogas causing higher BSFC in dual fuel mode operation. Moreover, due to raw biogas containing a higher percentage of non-combustible components, there is a reduction in fuel quality. In other words, due to a lower percentage concentration of methane in raw biogas leads to having lower calorific value. Thus, lower calorific values of biogas lead to having higher BSFC in biogas-diesel mode as compared to baseline operation. On the other hand, in dual-fuel mode, an increase in biogas concentration leads to step up brake-specific fuel consumption. This is due to increasing the amount of biogas concentration further minimizing the amount of injected pilot fuel. This decreases the energy content of the mixture and causes higher BSFC as an increment of biogas flow. An average BSFC increment of the biogas-diesel mode of D + BG@2 L/min, D + BG@4 L/min, and D + BG@6 L/min flow rate were obtained as 11.81, 16.43, and 20.87%, respectively, as compared to diesel mode.

*4.3. Emission Characteristics*

a.       Exhaust Emissions of Carbon Monoxide (% Vol.)

Carbon monoxide is produced as a result of incomplete combustion due to higher fuel proportion than stoichiometric during combustion with a deficiency of oxygen. On the other hand, it may also occur at lower combustion temperature in the case of the fuel

proportion in the mixture being much less than stoichiometric [27]. CO emission is subject to the availability of oxygen content, amount of carbon in the fuel, and the ability of a fuel to burn. For the duration of incineration of fuel, if there is less available oxygen, then the fuel cannot be completely oxidized and hence results in more CO emission due to the presence of carbon. Figure 7 illustrates the variation of carbon monoxide ingredients with respect to load for diesel fuel and diesel with biogas mixtures. In biogas-diesel operation, higher CO is observed than baseline. This is due to the reason of the lower flame speed of raw biogas because it consists of more $CO_2$, reduction of oxygen flow due to the addition of biogas, and higher specific heat of biogas, as compared to diesel, and biogas cannot completely combust at a temperature at which baseline combusts. The above reasons cause some portion of a mixture to proceed with complete combustion and hence results in more CO being released. It also increases as an increment of biogas flow, because high biogas flow further increases $CO_2$ concentration and decreases the availability of $O_2$ in the combustion chamber. Generally, as illustrated in Figure 7, an increase in engine load from 10 to 40% results in a decrease in CO emissions due to an increase in temperature and combustion efficiency. Further increase of load to 80 percent, CO emissions rise due to a decrease in the excess of oxygen caused by an increase in diesel fuel supply. CO emissions increase as the biogas feed rate increases due to a decrease in excess oxygen caused by replacing the inlet air with biogas supplied at the inlet. The average CO emission increment of D + BG@2 L/min, D + BG@4 L/min, and D + BG@6 L/min flow rate was 1.05, 11.57, and 21.40%, respectively, as compared with that of diesel mode.

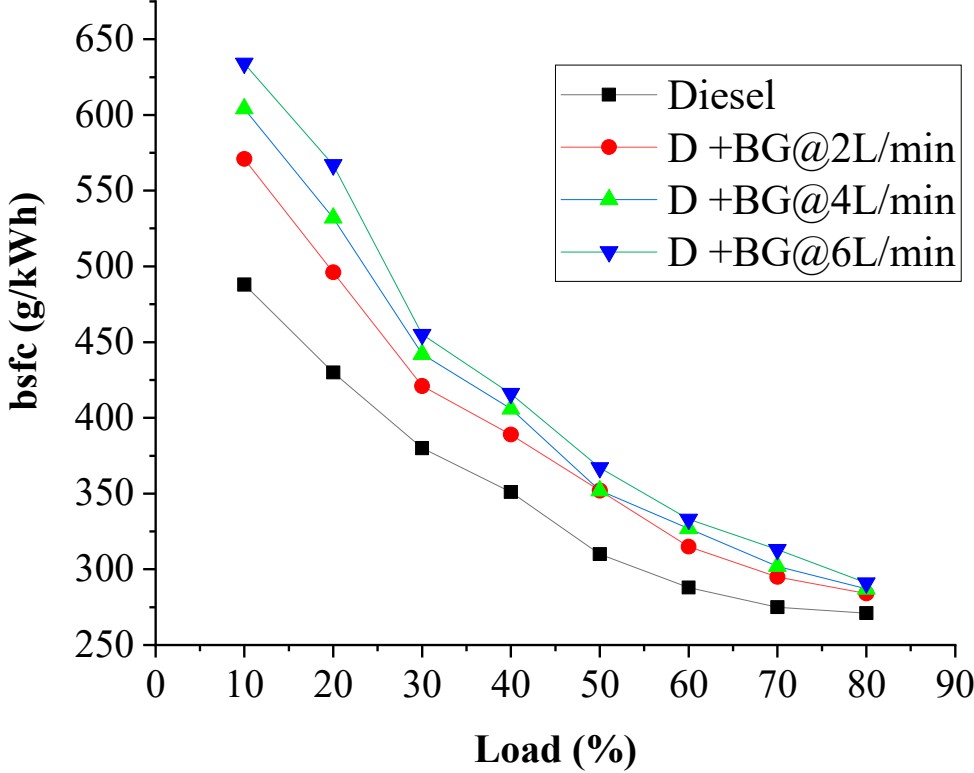

**Figure 6.** Variation of bsfc with respect to engine load.

b       Emissions of Carbon Dioxide

The $CO_2$ emission increases with increases in load for all operations as shown in Figure 8 below. This is because of an increase in combustion rate due to relatively high combustion temperature as compared to low loads. Hence, this higher combustion temperature at higher engine load provides the conversion of some portion of carbon monoxide into $CO_2$ and causes an increase in $CO_2$ emission. On the other hand, as raw biogas con-

tains a larger percentage of $CO_2$, an increase in the amount of $CO_2$ emission was noticed in biogas-diesel as compared to baseline. Additionally, an increment in the amount of biogas flow leads to an increase in biogas energy contribution, which causes an increase in carbon dioxide content. The average $CO_2$ emission increment of D + BG@2 L/min, D + BG@4 L/min, and D + BG@6 L/min flow rate from diesel mode was 12.8, 25.98, and 47.33%, respectively.

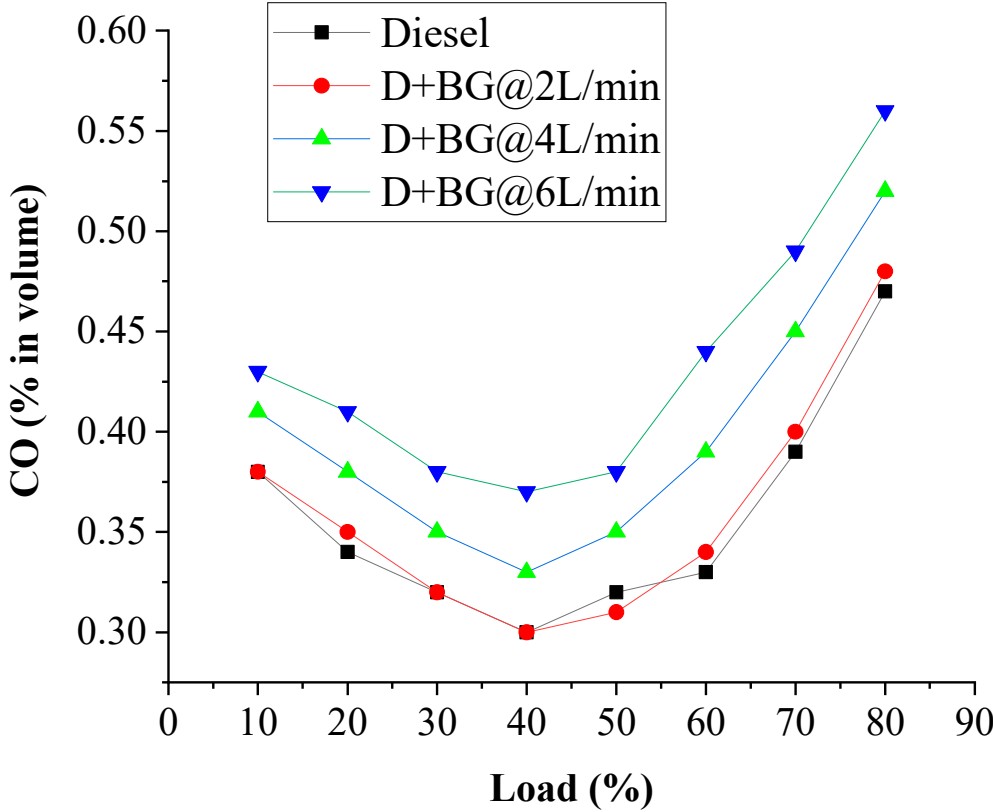

**Figure 7.** Variation of CO with respect to load.

c        Exhaust Emissions of Hydrocarbons

The hydrocarbon constituent released from the engine for both modes of operation with engine load is illustrated in Figure 9 below. The amount of unburn hydrocarbon (UHC) in biogas-diesel mode is lowered at 2 L/min and higher at 4 L/min and 6 L/min flow rates than baseline, throughout all tests. This is because of the poor burning of the dual-fuel caused by lower flame velocity of gaseous fuel, which contains unburnt components like $CO_2$, and reduction of oxygen concentration at higher biogas flow rates as paralleled to the baseline. Additionally, higher UHC in biogas-diesel operation at a higher flow of gas fuel is due to; longer ignition cunctation of dual-fuel, increase in BGES, which causes a reduction in combustion temperature; and carbon dioxide presence in biogas sucks up the heat and decreases the cylinder temperature, which decelerates the hydrocarbon oxidation process. Moreover, the overlapping between intake and exhaust valves for promoting dispelling of burned gasses after combustion is also a cause for unburnt HC to be incremented in case of feeding biogas-diesel fuel. Furthermore, unburnt HC increases as the quantity of gas fuel flow increases, this is because as the gas fuel flow increases, the amount of air entering the combustion chamber decreases, which results in a rich mixture as compared to a low biogas flow rate and causes incomplete combustion inside the cylinder. A similar trend was reported by [28]. In the same way, as CO does, an increase in engine load from 10 to 40% results in a decrease in HC emissions due to an increase in temperature and combustion efficiency. With a further increase of load to 80%, unburnt HC emissions rise due to a decrease in the excess of oxygen caused by an increase in diesel fuel supply. Unburnt HC

emissions increase as the biogas feed rate increases due to a decrease in excess oxygen caused by replacing the inlet air with biogas supplied at the inlet. The average unburnt HC emission reduction of D + BG@2 L/min was 0.52%% and increment of D + BG@4 L/min, and D + BG@6 L/min flow rate as compared with diesel mode was obtained as 3.26 and 6.54%, respectively.

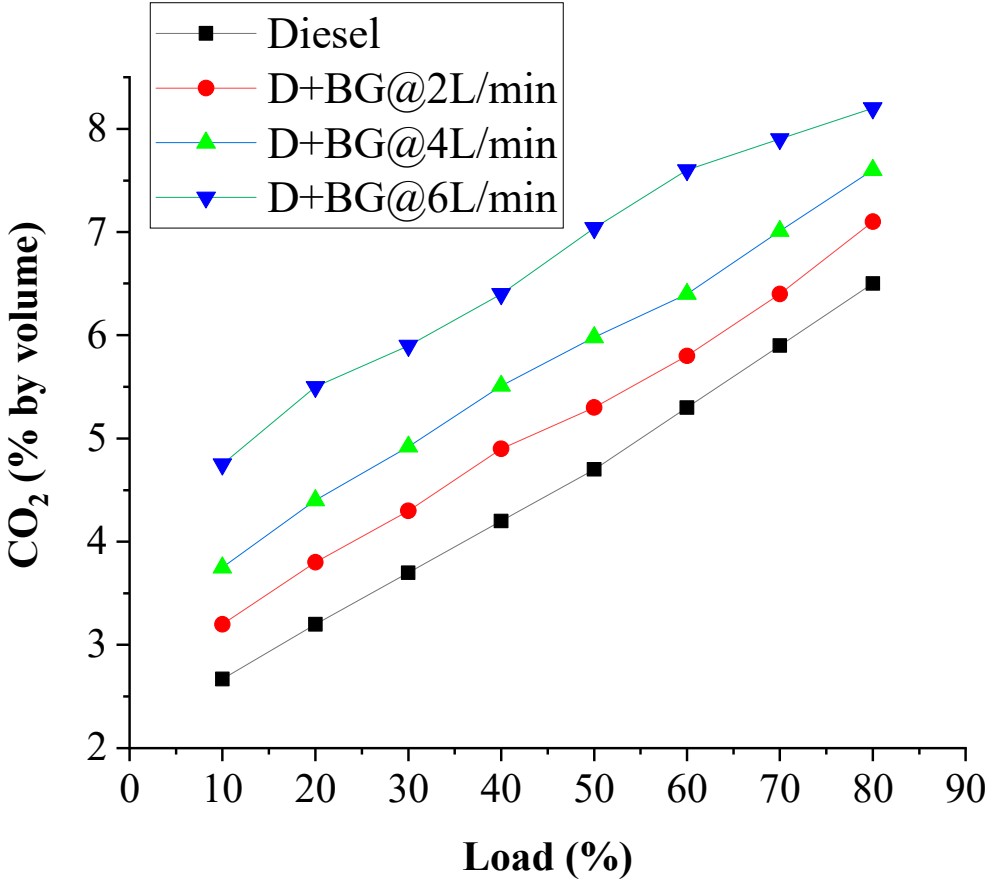

**Figure 8.** Variation of $CO_2$ with respect to load.

d  Nitrogen Oxide Exhaust Emissions (ppm Vol.)

The development of NOx emission is mostly subject to the availability of oxygen, higher temperature developed during the combustion, and the average time in which the occurrence of oxygen-nitrogen reactions happens up to completion, Bouguessa et al. [20]. The deviations of nitrogen oxide emission for baseline and dual-fuel mode are presented in Figure 10 below. From Figure 10, it is observed that NOx emission is lower in biogas-diesel mode as compared to that of the baseline fuel for all loads. This happens as a result of an increase in the equivalence ratio for the higher loads. In addition to this, the addition of biogas reduces the amount of oxygen introduced to the combustion chamber and this lowers the combustion temperature so that all the oxygen available in the combustion chamber is not employed to burn the fuel and the combustion becomes incomplete, this factor reduces the formation of NOx. Moreover, an increment in the biogas energy contribution leads to lower peak cylinder temperature in the mixture of both fuels and provides a lower NOx level (i.e., that increasing the share of biogas leads to a decrease in local maximum temperatures, and, accordingly, nitrogen oxides). Overall, an increment in the quantity of NOx constituent was noticed with an increment of engine load for all operations. The average NOx emission reduction of D + BG@2 L/min, D + BG@4 L/min, and D + BG@6 L/min flow rate as compared from diesel mode were obtained 19.91, 27.33, and 39.16%, respectively.

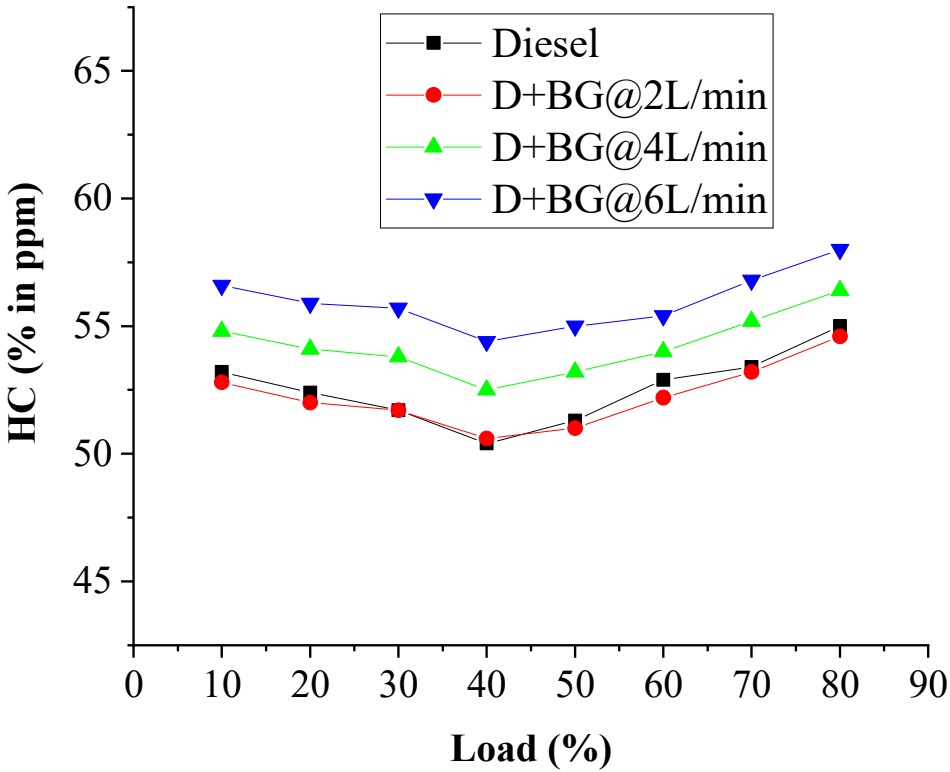

**Figure 9.** Variation of HC with respect to load.

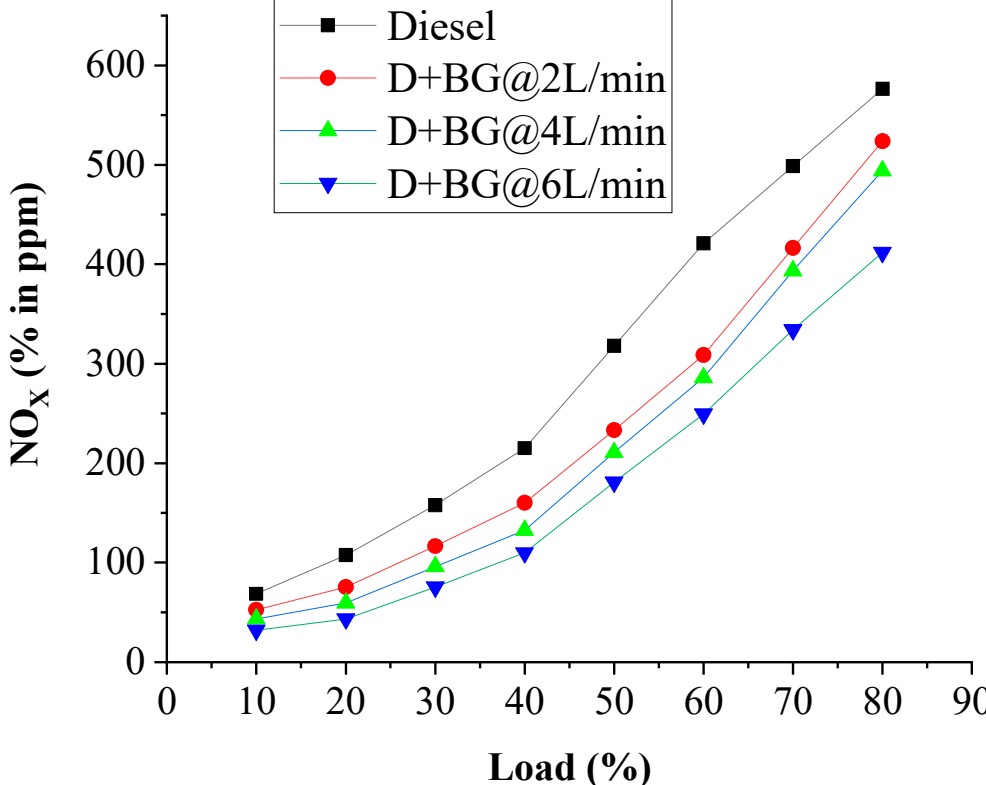

**Figure 10.** Variation of NO$_X$ with respect to load.

### 5. Conclusions

The performance parameters and emission constituents generated by a diesel engine running on biogas-diesel dual-fuel with changing engine load and gas fuel flow rate at a

constant engine speed of 1500 RPM were determined using an air-biogas mixing device in the air intake system. The following conclusions are drawn from the experiments carried out in the biogas-diesel dual-fuel mode diesel engine under various load conditions with variable gas fuel flow at constant engine speed, and thus from the results obtained:

As biogas flow rate increases from 2 to 6 L/min with an increase of load from 10 to 80%:

✓ There is an increase in average biogas energy share of 24.98 to 52.32% and diesel fuel replacement ratio of 13.41 to 23.29%.
✓ Relatively reduction in BTE from 11.19 to 25.72%, and an increment of BSFC from 11.81 to 20.87%.
✓ Increment in emissions like CO by 1.05 to 21.40%, $CO_2$ by 12.8 to 47.33%.
✓ Reduction of HC by 0.52% at 2 L/min flow rate and increments by 3.26 and 6.54% at four and six L/min flow rates, respectively.
✓ There is a reduction in NOx emission from 19.91 to 39.16%, respectively.

Generally, among different biogas-diesel operations, as a result of a small percentage of methane in given biogas, a biogas flow rate of 2 L/min yields relatively higher performance as compared to other flow rates but lower than diesel. On the other hand, higher CO and $CO_2$ emission characteristics but lower HC emissions at 2 L/min flow rate and allow extremely low levels of NOx as compared to diesel fuel operation.

As stated in the introduction, using a renewable alternative fuel, biogas, for the IC engine has several advantages. However, some changes and improvements could be made and improved in the future from this standpoint.

i. Purification of biogas: Biogas must be purified before it can be used as a fuel in internal combustion engines. As a result, only methane is combustible among the various components of biogas. Because raw biogas contains less methane and a high amount of incombustible gas such as carbon dioxide, the in-cylinder temperature drops, allowing incomplete combustion to occur. Purified biogas, on the other hand, contains more than 90% methane content. As a result, when compared to diesel fuel, the use of purified biogas in CI engines results in improved performance and lower emissions.

ii. Venturi mixer optimization: Even if the CI engine is supplied with purified biogas, a poorly mixed air–biogas mixture has a negative impact on performance and emission characteristics. All of the parameters of the venturi air-biogas mixer should be optimized using ANSYS fluent to achieve better mixing quality and, as a result, a more homogeneous air–biogas mixture.

**Author Contributions:** Conceptualization, M.G.L. and M.W.M.; methodology, M.G.L. and M.W.M.; investigation, M.G.L. and M.W.M.; resources, M.G.L. and M.W.M.; data curation, M.G.L.; writing—original draft preparation, M.G.L.; writing—review and editing, M.G.L. and M.W.M.; visualization, M.G.L.; supervision, M.W.M.; project administration, M.G.L. and M.W; funding acquisition, M.G.L. All authors have read and agreed to the published version of the manuscript.

**Funding:** The authors would like to acknowledge Adama Science and Technology University, Ethiopia, for financial support with the research grant number of ASTU/SM-R/110/19.

**Institutional Review Board Statement:** Not applicable.

**Informed Consent Statement:** Not applicable.

**Data Availability Statement:** The dataset used in this research are available upon request from the corresponding author.

**Conflicts of Interest:** The authors declare no conflict of interest.

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
