# Peer review of "Investigation of the Performance and Emission Characteristics of Diesel Engine Fueled with Biogas-Diesel Dual Fuel"

_2673-3994, doi:10.3390/fuels3010002_

Round 1

Reviewer 1 Report

In my opinion, authors should add a new idea to this type of study.  Is this approach feasible and sustainable for large-scale investigations?  I cannot recommend the publication of this investigation in the present form. The novelty and potential impact of this paper should be clarified.  I recommend that you edit the entire article according to journal standards and re-send it. An article that is not well prepared, is more a research technical report.

Avoid real pictures, but use only schematic pictures for the test bench.

Please for you reference have a look to some research paper that can be used for your reference for the next submission and as guidelines: 10.1007/978-981-15-0418-1_8, 10.1007/978-981-16-0931-2_2, 10.1016/j.energy.2014.08.059. The current format of the paper is not acceptable to me. They should be able to propose a model to cover a gap based on the literature and they should evaluate the model based on justifications and comparisons concerning those proposed in the literature. Finally, the readers of this paper should be able to get the main points through some numerical results and summary.

Reviewer 2 Report

This paper examines combustion of biogas /diesel blend in an engine. This paper needs major revisions before considering for publication. The clarification of the following points would help to improve the paper.

  1. This manuscript discusses a widely investigated topic. Most of the trends presented are generally known and there is only very little new except a new set of experimental data. The novelty of the work must be clearly addressed and discussed. Research gap should be delivered in more clear way with directed necessity for the conducted research work.
  2. Research method: The selection of a specific blending ratio and engine operative conditions is not sufficiently justified. The applied methodologies and measurements procedure are weakly described.
  3. Accuracy of measurement is not adequately discussed; error bars should be reported in all the plots and graphs.
  4. Fig 5-10 show that overall engine performances for biogas/diesel are poorer than diesel. Is biogas/diesel a viable replacement for diesel? Please clarify.
  5. Discussion in section 4 is lacking references. Please cite relevant papers if theories used for explanation belong to other authors.
  6. Conclusion section is missing some perspective related to the future research work and cleaner production aspect must be addressed. Please enrich the conclusion section, demonstrating the innovation and main outcomes of the conducted research.

Reviewer 3 Report

In my opinion, the article “Investigation of the Performance and Emission Characteristics of Diesel Engine Fueled with Biogas-Diesel Dual Fuel” is of some interest. It describes the method of obtaining and the composition of biogas, an experimental installation, a test procedure and analyzes the results of experimental studies. A feature of the article is the use of biogas, which, in addition to methane, includes oxygen, carbon dioxide and other impurities.

Below are the main notes on the content of the article.

  1. The main purpose of the work is not formulated in the article.
  2. In the text of article, references to literature are not drawn up according to the rules. They should be corrected. Reformat the “References” section at the same time.
  3. In Table 1, the total composition of raw biogas components is only 99.5%. There is probably a mistake somewhere. Water content is indicated in “ppm”, but should be given as a percentage.
  4. In the explanations to Table 1, I recommend to indicate the approximate composition of impurities. This would be useful in assessing the heat capacity of biogas.
  5. On page 3, the text after the title of figure 1 should start on a new line.
  6. In the same place the title of the third section “3. Experimental Setup” should also start on a new line.
  7. The article does not indicate the measurement errors of the used gas analyzer.
  8. Figure 3 incorrectly indicates that all exhaust gases are passing through the gas analyzer. In fact, only a small fraction of the exhaust gas is supplied to the gas analyzer.
  9. The “Methodology” section is very short. It should be described in more detail and supplemented with information on the modes of supply of biogas and a device for measuring the rate of its supply.
  10. The description for “Experimental Setup” on page 3 indicates that the unit is liquid cooled. And in table 3 the air cooling of the engine is recorded. There is probably a mistake somewhere.
  11. In Table 3, the right column name is erroneously spelled “TM3-02”. It should be renamed, for example to “Value”.
  12. The simultaneous use of gaseous and liquid fuels with different characteristics makes it difficult to quantify BTE, BSFC and BGES. Therefore, I recommend at the beginning of section “4. Result and Discussion” to provide expressions for the definition of these parameters. Next, show the change in the mass fraction of biogas depending on the load at different rates of biogas supply. After that, bring the change in the share of biogas energy from the load (Figure 7), the change in BTE (Figure 5) and the change in BSFC (Figure 6). This arrangement of the sections will make it possible to more logically explain the obtained results of changes in BTE and BSFC with references to the previously given change in the mass fraction of biogas in the fuel mixture.
  13. On page 6, the title of the section “4.3 Emission characteristics” should start on a new line.
  14. In sections 4.3 (a) and 4.3 (c), I recommend that the authors add the following clarifications. 1) With an increase in engine load from 10% to 40%, a decrease in CO and HC emissions occurs as a result of an increase in temperature and combustion efficiency. 2) With a further increase in the load to 80%, the emissions of CO and HC increase due to a decrease in the excess of oxygen as a result of an increase in the supply of diesel fuel. 3) At a higher biogas feed rate, CO and HC emissions also increase due to a decrease in excess oxygen as a result of replacing the inlet air with biogas supplied at the inlet.
  15. In section 4.3 (d), I recommend that the authors add that increasing the share of biogas (that is, replacing diesel fuel with biogas) leads to a decrease in local maximum temperatures and, accordingly, nitrogen oxides.

In my opinion, the article has scientific novelty and, after appropriate corrections, can be published in the journal “Fuels”.

Round 2

Reviewer 1 Report

I cannot recommend it for publication the authors need to improve the quality and scientific soundness of this manuscript. The reviewer has reported some open remarks in the previous review report but the authors have not applied them. For example, remove the real picture of the laboratory, just include some scheme. Some papers as references were suggested not to cite it but to have an example to improve the scientific content of your manuscript. I regret to inform you that I cannot recommend it for publication.

Reviewer 2 Report

Comments were addressed adequately. 

Author Response

Thanks for your comments